# In vivo structure and dynamics of the SARS-CoV-2 RNA genome

Yan Zhang [1,6], Kun Huang[2,6], Dejian Xie[3,6], Jian You Lau [4], Wenlong Shen[1], Ping Li [1], Dong Wang[5], Zhong Zou[2], Shu Shi[1], Hongguang Ren[1], Youliang Wang [1], Youzhi Mao[3], Meilin Jin[2✉], Grzegorz Kudla [4✉] & Zhihu Zhao [1✉]

The dynamics of SARS-CoV-2 RNA structure and their functional relevance are largely unknown. Here we develop a simplified SPLASH assay and comprehensively map the in vivo RNA-RNA interactome of SARS-CoV-2 genome across viral life cycle. We report canonical and alternative structures including 5'-UTR and 3'-UTR, frameshifting element (FSE) pseudoknot and genome cyclization in both cells and virions. We provide direct evidence of interactions between Transcription Regulating Sequences, which facilitate discontinuous transcription. In addition, we reveal alternative short and long distance arches around FSE. More importantly, we find that within virions, while SARS-CoV-2 genome RNA undergoes intensive compaction, genome domains remain stable but with strengthened demarcation of local domains and weakened global cyclization. Taken together, our analysis reveals the structural basis for the regulation of replication, discontinuous transcription and translational frameshifting, the alternative conformations and the maintenance of global genome organization during the whole life cycle of SARS-CoV-2, which we anticipate will help develop better antiviral strategies.

[1] Beijing institute of Biotechnology, Beijing, China. [2] Unit of Animal Infectious Diseases, National Key Laboratory of Agricultural Microbiology, College of Veterinary Medicine, Huazhong Agricultural University, Wuhan 430070, China. [3] Wuhan Frasergen Bioinformatics Co., Ltd, Wuhan, China. [4] MRC Human Genetics Unit, University of Edinburgh, Edinburgh EH4 2XU, UK. [5] Department of Microbiology, University of Hong Kong, Queen Mary Hospital, Pokfulam, Hong Kong, China. [6] These authors contributed equally: Yan Zhang, Kun Huang, Dejian Xie. ✉email: jinmeilin@mail.hzau.edu.cn; gkudla@gmail.com; zhaozh@bmi.ac.cn

The coronavirus disease 2019 (COVID-19) pandemic, caused by the severe acute respiratory syndrome coronavirus 2 (SARS-CoV-2) coronavirus, has caused more than three million deaths worldwide at the time of submission. Although many efforts have been devoted to control the disease and vaccines are being approved for emergency use, the pandemic is far from being under control. Therefore, there is an urgent need to understand the basic molecular biology of SARS-CoV-2 coronavirus. SARS-CoV-2 belongs to the broad family of coronaviruses. It is a positive-sense, single-stranded RNA virus, with a single linear RNA segment of ~30,000 bases[1].

Coronavirus RNA-dependent RNA synthesis includes two different processes as follows: continuous genome replication that yields multiple copies of genomic RNA (gRNA) and discontinuous transcription of a collection of subgenomic mRNAs (sgRNAs or sgmRNAs), which encode the viral structural and accessory proteins[2,3]. The transcription process is controlled by transcription-regulating sequences (TRSs) located at the 3′-end of the leader sequence (TRS-L) and preceding each viral gene (TRS-B), and requires base pairing between the core sequence of TRS-L and the nascent minus strand complementary to each CS-B (cCS-B), allowing for leader-body joining[2,4,5]. A three-step working model of coronavirus transcription was suggested[2,6], which implies that long-distance RNA–RNA interactions are required prior to template switch. Long-distance interactions between B motif (B-M) and its complementary motif, and between proximal element and distal element, are important for forming high-order structures promoting discontinuous RNA synthesis during N sgmRNA transcription in the Transmissible Gastroenteritis Virus (TGEV) (see review[5] for details). However, the motifs involved in these interactions are not conserved in β-coronaviruses and it is not known if similar interactions contribute to transcription of other sgmRNAs. Therefore, although it is widely assumed that TRS-L interacts with cCS-B, there has been no experimental evidence for direct interactions between TRS-L and TRS-Bs.

Functional studies have revealed the importance of RNA secondary structures for viral replication, transcription, and translation[7–9]. One unique feature of the coronavirus is frameshifting in ORF1ab, giving rise to RNA-dependent RNA polymerase (RdRP) and other proteins in ORF1b. The structure of the SARS-CoV frameshift element (FSE; whose sequence differs from the SARS-CoV-2 FSE by just one nucleotide) was solved by nuclear magnetic resonance (NMR) to be a three-stem pseudoknot[10] and was proposed to play key roles in translational control of ORF1b[11]. The global SARS-CoV-2 RNA structure in cells was also proposed by the genome-wide chemical probing strategies[12–15]. Recently, alternative conformations of the FSE were derived from in-cell SHAPE/DMS-MaP seq data[14,15]. Although these methods provided valuable insights into *cis*-acting RNA structures regulating important biological processes of virus life cycle, they could not elucidate long-distance interactions. In addition, short- and long-distance interactions within the SARS-CoV-2 RNA were described using the COMRADES and vRIC-seq methods. Importantly, Ziv et al.[16] discovered networks of both gRNA and subgenomic RNA (sgRNA) interactions by applying specific probes to pull down each RNA species in cells, and Cao et al.[17] reconstructed structures in virions. However, all of these experiments were performed in a specific stage of the virus life cycle. Thus, there is need to directly compare structures from different stages to investigate their dynamics and functional relevance during the whole life cycle.

In this study, to comprehensively map RNA–RNA interactions of SARS-CoV-2 RNA both in cells and in virions, we used a simplified sequencing of psoralen crosslinked, ligated, and selected hybrids (SPLASH)[18] approach. We provide direct experimental evidence of comprehensive TRS-L interactions with TRS-

B regions of sgRNAs, and identify and validate novel sgRNAs by analyzing additional TRS-L interaction peaks. We find multiple alternative interactions mediated by FSE, providing structural basis for ribosome stalling. In addition, we show that both proximal and distal genome RNA–RNA interactions are strengthened, whereas sgRNA-mediated interactions are significantly reduced in virions, suggesting thorough compaction of genome RNA in virions. Interestingly, although TRS-L-mediated interactions including genome cyclization are weakened, interactions between TRS-L and open reading frame ORF S are strengthened in later phase of infection cells and mature virions, which may contribute to rapid transcription of sgRNAs. Our data provides a global landscape of relationships between SARS-CoV-2 RNA structure and key processes of virus life cycle, such as replication, discontinuous transcription, and translation.

## Results

### Overview of short- and long-range RNA–RNA interactions of SARS-Cov-2.
We developed a simplified SPLASH protocol to capture RNA–RNA interactions in the SARS-CoV-2 virus. Briefly, the biological samples are first stabilized by Psoralen-PEG3-Biotin crosslinking, followed by RNase III treatment, proximity ligation, library preparation, and high-throughput sequencing. Samples were collected from different phases of the SARS-CoV-2 virus life cycle to infer the dynamic structure of viral RNA at different stages. In the early stage of infection, we collected virus-infected Vero cells (C), in which cytopathic effect (CPE) was not observed. At a later stage, when 70% of cells underwent CPE, we collected the cell culture supernatant and harvested the mature virus particles (V), and at the same time used freeze–thaw methods to lyse the cells (L) to collect the cell and virus RNA (Fig. 1a). The major steps are shown in Fig. 1a. Pearson's correlation analysis of chimera counts between biological replicates indicated high reproducibly of experiments (Fig. 1b and Supplementary Figs. 1 and 2a, b). Chimeric signals were significantly higher in ligated than in non-ligated samples (Supplementary Fig. 1 and Supplementary Data 1). As documented by others[19,20], we also observed weak chimeric signals in non-ligated samples. Counts of chimeras in ligated and non-ligated samples were correlated, particularly in the C sample (Supplementary Fig. 2c–e), consistent with the idea that chimeras in non-ligated samples may be due to endogenous ligation activity derived from host cells. The ligated RNA fragments could form both 5′–3′ and 3′–5′ chimeras[21] (Fig. 1c, d). Interestingly, we noticed there are more 3′–5′ chimeras than 5′–3′ chimeras, especially when two arms of chimeras are in short distance (Supplementary Fig. 3a). This is also true for recent COMRADES data[16] (Supplementary Fig. 3b), possibly because local 3′–5′ chimeras can be easily detected bioinformatically, whereas local 5′–3′ chimeras are difficult to distinguish from nonchimeric reads. The contact matrices are provided as Supplementary Data 2. Motif analysis of the ends of chimeras showed no enrichment of GG dinucleotides or GGG trinucleotides, suggesting that chimeras do not result from template switching.

### Structures of UTRs and genome cyclization.
The 5′-untranslated regions (UTRs) of coronaviruses contain five evolutionarily conserved stem-loop structures (denoted SL1–SL5), which are essential for genome replication and discontinuous transcription[8,22,23]. In our data, all five stems and recently identified stem loops (SL6, SL7, and pSL8)[12,14,24] are supported by chimeras (Supplementary Fig. 4a). To resolve base pairing at single-nucleotide resolution, we calculated base-pairing scores for all candidate base pairs along the virus genome, analogous to COMRADES scores as described in ref. [25]. Notably, there is a small stem loop after SL4 according to

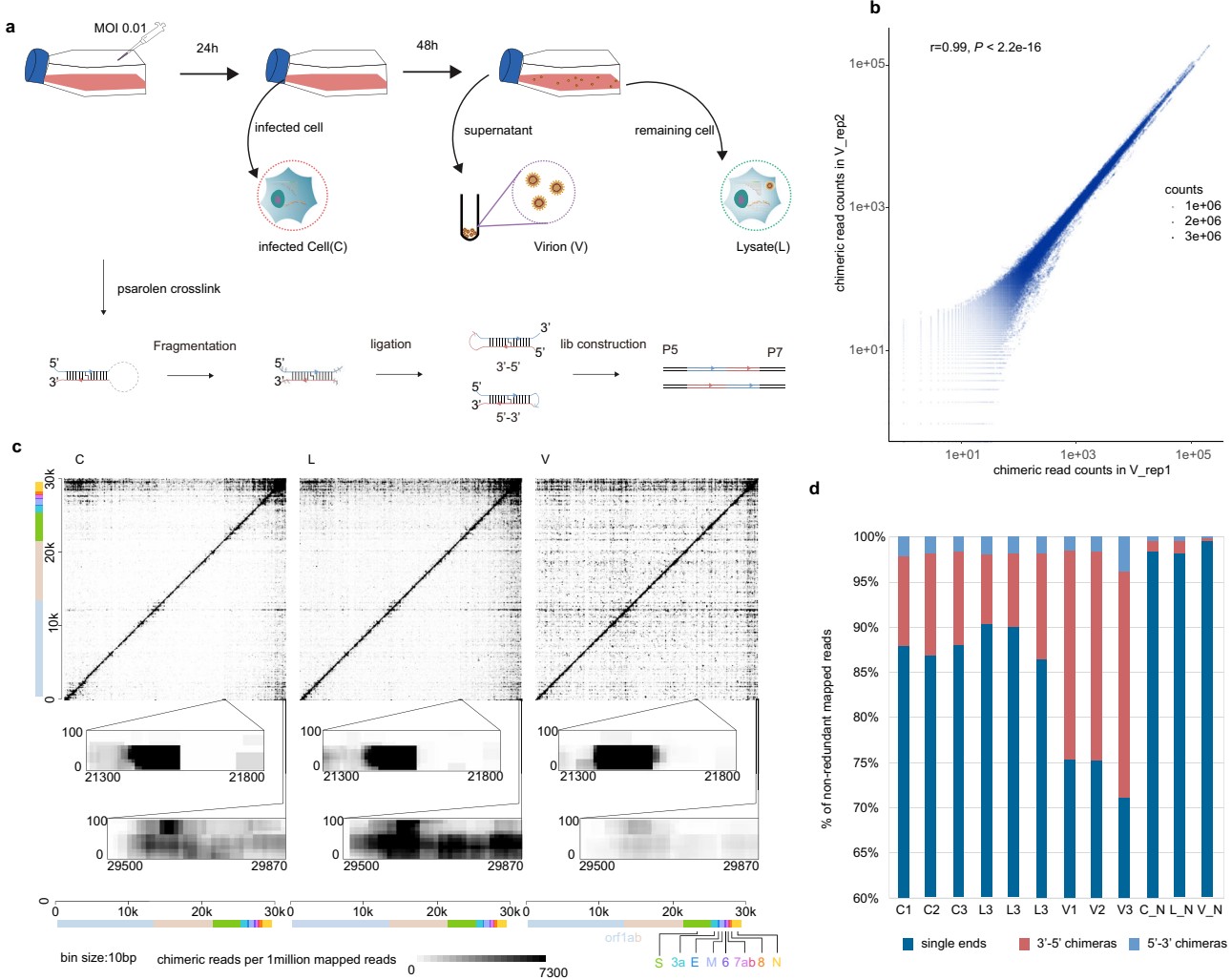

**Fig. 1 Overview of the experiment. a** Schematic diagram for sample collection and major experimental steps. **b** Dotplot shows chimeric read counts from two replicates, indicating good reproducibility of proximity protocol. r: two-sided Pearson's correlation coefficient, P: Pearson's correlation test *P*-value. **c** Heatmap of RNA–RNA interactions along the SARS-CoV-2 gRNA. Each dot represents an interaction signal between the genomic coordinates on the *x* and *y* axes. *X* axis shows the coordinates of the 5′-arm of the chimera and the *Y* axis shows the 3′-arm of the chimera. So, 5′–3′ chimeras are above the diagonal and 3′–5′ chimeras are below the diagonal. Zoomed inset plots show TRS-L–S interaction and genome cyclization. **d** Statistics of mapped single-end RNA, 3′–5′ chimeras, and 5′–3′ chimeras in each sample.

computational prediction[26]; however, most icSHAPE and related methods and COMRADES do not support this stem loop[12–16,27]. Similarly, our data does not support the existence of this stem loop (gray dashed rectangle in Supplementary Fig. 4b–d). Given that 5′-UTR structure was reported to be similar in several recent publications, we used this structure as a gold standard to test sensitivity and specificity of our method to detect local structure. Receiver operating characteristic (ROC) curves showed accurate detection of the 5′-UTR structure in our data (Supplementary Fig. 4e, f). Contact matrices are similar among C, L, and V samples, indicating similar structures in 5′-UTR during virus life cycle. On the other hand, SL1 ~ SL3 are weaker in cells than in virions (comparing normalized base-pairing scores by paired Wilcoxon's rank-sum test, *P* < 0.001), possibly indicating that these stem loops form more alternative structures in cells (i.e., TRS-L : TRS-B or genome cyclization, which will be discussed below).

The 3′-UTR contains a bulged stem loop (BSL), hypervariable region comprising a conserved octonucleotide sequence and the stem-loop II-like motif (S2M), which are essential for sgRNA synthesis in mouse hepatitis virus (MHV)[7,28,29]. All of these structures could be detected in our data (Supplementary Fig. 5a,

b). Specifically, there are base-pairing chimeras in the pseudoknot in every stage (Supplementary Fig. 5b), suggesting the existence of this pseudoknot. These structures are also confirmed by recent three-dimensional (3D) modeling from multiple data[30]. Interestingly, there are also chimeras supporting this structure in COMRADES data, but with much lower coverage. Due to the relative low coverage, we speculate that this pseudoknot is not predominant, as proposed[30,31]. In addition, recent deconvolution of SARS-CoV-2 RNA structure suggested more conformations (conformation B) of 3′-UTR. We also evaluated base-pairing scores of this conformation and found BSL and P2 are supported, indicating bona fide alternative structure of a 3′-UTR (Supplementary Fig. 5c). At the same time, two stems of conformation B have no base-pairing chimeras (gray dashed rectangles in Supplementary Fig. 5c), whereas S2M had a strong signal (Supplementary Fig. 5a). Therefore, we identified an alternative 3′-UTR structure in the S2M region (Supplementary Fig. 5d), similar to a recently reported structure[13]. We also checked these structures in COMRADES data and found almost the same pattern in base pairing (Supplementary Fig. 5e, f). By comparing base pairing, we found base-pairing scores were lower in virions

(V) than in cells (C) ($P = 1.72 \times 10^{-6}$, paired Wilcoxon's sum-rank test), indicating that the pseudoknot is weakened in virions. It is possible that conformation of 3′-UTR is complicated and dynamic, with a dominant canonical conformation (conformation A in ref. [32]), and alternative conformations, including conformation B in ref. [32] and alternative S2M.

In the contact matrices, we also found interactions formed by 5′-end and 3′-end of the SARS-CoV-2 genome, indicating genome cyclization (Fig. 1c and Supplementary Fig. 6), which was previously described[16]. Interestingly, we found additional base pairing at genome cyclization sites (Supplementary Fig. 6b), implying that SL4 in 5′-UTR is also involved in cyclization processes. Notably, genome cyclization is reduced in virions (Supplementary Fig. 6a, b). Genome cyclization was also described in other viruses, including flaviviruses[25], and involved in replicase recruitment at least in Dengue virus and Zika virus[9]. Considering dynamics of genome cyclization upon packaging and releasing, we speculate that genome cyclization is involved in replication and/or packaging. Perturbation of genome cyclization might offer an interesting avenue to target SARS-CoV-2 replication.

**Long-distance interactions between TRS-L and TRS-B regions**. Although RNA proximity ligation generates chimeric reads indicative of RNA–RNA interactions, cells also contain many spliced transcripts, or in the case of coronavirus-infected cells, sgRNAs, which resemble chimeras produced by proximity ligation, because they originate from disjoint regions of the genome. To correctly identify RNA–RNA interactions, it is therefore essential to filter chimeras that result from splicing or discontinuous transcription. In previous studies, filtering was performed by mapping reads to a database of known transcripts, and removing reads mapped to known splice junctions[33–35]. However, SARS-CoV-2 produces a diversity of sgRNAs[3], many of which are not yet annotated, rendering this approach impractical. Instead, we empirically assessed the characteristics of chimeric reads found in a published RNA sequencing (RNA-Seq) data set from SARS-CoV-2-infected cells[3], which we assumed to represent sgRNAs rather than RNA–RNA interactions.

We identified three characteristics that differentiated most sgRNA chimeras found in RNA-Seq from bona fide RNA–RNA interactions chimeras: (1) sgRNAs were ligated almost exclusively in the 5′–3′ orientation, whereas RNA proximity ligation chimeras can be ligated in both 5′–3′ and 3′–5′ orientation (reviewed in ref. [21]); (2) the junctions between the arms of chimeras were precisely localized in sgRNAs, whereas ligation sites in proximity ligation were variable, due to the random nuclease digestion step used in proximity ligation[36]; and (3) sgRNA chimeras typically included regions of homology between TRS-L and TRS-B sides of the chimera[37], whereas proximity ligation chimeras typically include no such regions[36]. Adjustment of the maximum gap/overlap setting in our analysis pipeline, hyb, allows detection (gmax = 20, "relaxed pipeline," allowing a maximum of 20 nt gap or overlap between chimera fragments) or removal (gmax = 4, "stringent pipeline," see "Methods" section for details) of most sgRNA chimeras in RNA-Seq data, whereas proximity ligation chimeras are detected with both settings (Supplementary Data 1).

Notably, although both 5′–3′ and 3′–5′ chimeras are detected with the stringent pipeline (Supplementary Fig. 7a), chimeras detected with the relaxed pipeline are almost all in 5′–3′ orientation (Supplementary Fig. 7b) and junction sites (defined as ligation points of the two fragments, Supplementary Fig. 7c) are highly localized (Supplementary Fig. 7d, e). We therefore used the stringent pipeline to analyze proximity ligation data, while filtering away contaminating sgRNAs.

To identify RNA–RNA interactions mediated by TRS-L, we applied a viewpoint analysis[25] to 5′–3′ and 3′–5′ chimeras. We found multiple TRS-L interaction peaks along the SARS-CoV-2 genome and these peaks were adjacent to the 5′-end of canonical sgRNA regions (Fig. 2a, b, d) and particularly obvious for 3′–5′ chimeras. By contrast, there were few 3′–5′ chimeras in non-ligated samples or RNA-Seq data (Supplementary Fig. 7a). To rule out the possibility that these 3′–5′ chimeras come from sgRNAs, we analyzed the distribution of junction sites of TRS-L and found that these sites were highly variable (Fig. 2c). Furthermore, the varied mapped positions of both arms of chimeric reads spanning junction sites, as shown in Fig. 2e and Supplementary Fig. 8 for examples, further supported the origin of chimeras from genome folding rather than sgRNA transcripts. Therefore, simplified SPLASH data contain both TRS-L-mediated RNA–RNA interactions and TRS-L-dependent sgRNAs, which can be discriminated by our methods. RNA base pairing mediated by long-range interaction indicated that TRS-L may stably associate with TRS-B regions (Fig. 2f and Supplementary Fig. 9). Interestingly, we noticed that TRS-L usually does not interact with the exact TRS-B sequence, but with flanking sequence within 50 nt away. This might provide flexibility for the next step of pairing to Ccs-B and template switching.

**Identification and validation of novel TRS-L-dependent sgRNAs**. Apart from canonical sgRNAs, we also observed additional regions interacting with TRS-L (black arrowhead indicated in Fig. 2a), with one of them (3.9 K) also identified in recent report by Ziv et al.[16]. The contact matrix based on 3′–5′ chimeric reads and an analysis of individual chimeras showed specific interactions (Supplementary Fig. 10). These regions form stable base pairing with the TRS-L region (Supplementary Fig. 10c, e). To check whether the TRS-L-mediated interactions give rise to candidate sgRNAs, we performed reverse-transcription PCR (RT-PCR) in independent non-crosslinked cells. Sanger sequencing results confirmed these sgRNAs indeed exist (Supplementary Fig. 10f, g). Interestingly, although these novel sgRNA have no canonical ACGAAC core sequence motif (CS-B), they both partially overlap with the canonical CS motif. This indicates that TRS-L and partial cCS-B base pairing at the negative strand are critical for template switching in discontinuous transcription of these sgRNAs.

It should be interesting to further validate the expression and function of the novel transcripts in the future. This analysis also emphasizes the value of our experiment in dissecting interaction and discontinuous transcription of coronaviruses.

**Canonical and alternative structure around FSEs**. A characteristic feature of coronaviruses is the programmed-1 ribosomal frameshifting to facilitate translation of ORF1b encoding RdRp and control the relative expression of their proteins. A three-stem pseudoknot structure was proposed[11] and was recently proven by cryo-electron microscopy[38] and NMR[31]. Therefore, we sought to analyze both local and long-range interactions around FSE.

In our data, the proposed three-stem pseudoknot[11] was supported by chimeric reads (Fig. 3a, b), whereas at the same time we also found alternative local structures embedding the FSE in a larger stable stem loop (arch1 in Fig. 3a, b), which are all supported by chimeras (Fig. 3b, alternative local structure1). Specifically, we also evaluated base pairings in FSE structures as proposed recently[12,14] and found all the stems are supported by chimeras (Fig. 3b). Interestingly, base pairing remains throughout the virus life cycle and scores are higher in virions (Supplementary Fig. 11a). Canonical pseudoknot interactions around FSE are also supported by COMRADES data[16] (Supplementary Fig. 11b).

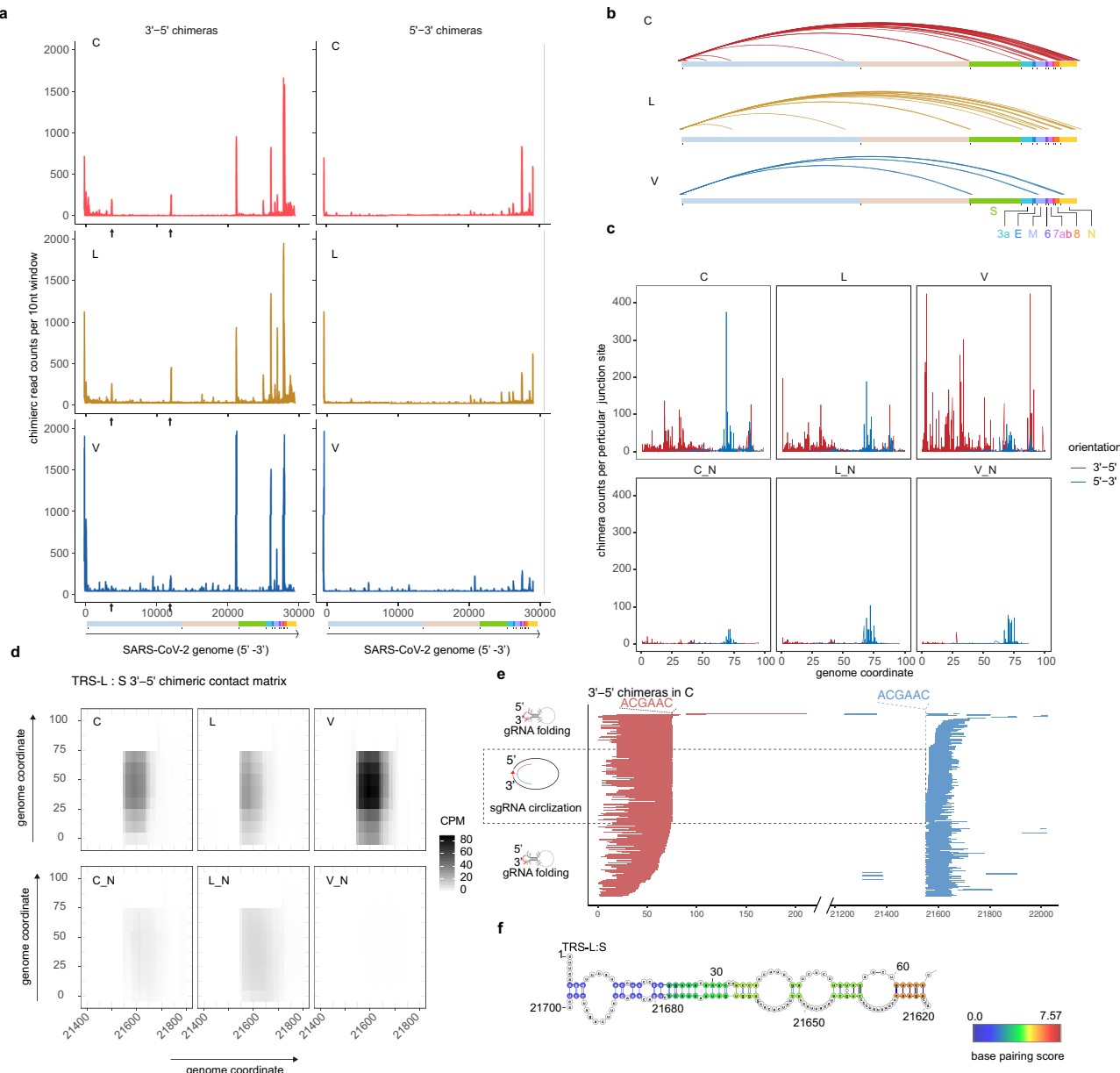

**Fig. 2 TRS-L interact with canonical TRS-B sites. a** Viewpoint histograms showing binding positions of the TRS-L region (first 100 nt) along the SARS-CoV-2 genome in indicated samples. The 3′–5′ chimeras and 5′–3′ chimeras were separately plotted. Black arrowheads indicate additional peaks in orf1a. **b** Enriched TRS-L interaction peaks deduced from *Z*-score method. Chimeric read counts from bin–bin contacts were normalized by *Z*-score, then interactions with *Z*-score > 2.13 (95% confidence of being above average) and mediated by TRS-L were plotted. **c** Junction site distribution on TRS-L region (the first 100 nt), chimeras that break at exactly particular base were counted, showing that the ligation happened in varied sites. Lines show frequencies of junction points of chimeras (3′–5′ chimeras in red and 5′–3′ chimeras in blue), which localized before nt100. **d** Contact matrix of 3′–5′ chimeric reads spanning TRS-L: S junction sites. Color depicts counts of chimeric reads per one million mapped reads (CPM). **e** Randomly selected 3′–5′ chimeras overlapping the TRS-L: S junction sites. The red lines indicate 3′-arms of chimeric reads, whereas blue lines indicate 5′-arms of chimeric reads. Chimeric reads with varied ends are derived from random fragmentation and ligation, reflecting long-range RNA–RNA interactions. **f** RNA base pairing between TRS-L and upstream of S, paired bases were colored by log2 chimeric read counts supporting each base pair (in C sample).

Surprisingly, we also found several alternative long-range interactions mediated by FSE (Fig. 3a). Besides FSE-arch proposed by Ziv et al.[16] (referred to as Ziv's arch hereafter), the alternative arches are formed by FSE region and upstream ~620 nt (arch 2) and ~1.1 kb (arch 3) elements, respectively (Fig. 3a). These elements form stable base pairing with the FSE region (Fig. 3b, c and Supplementary Fig. 12). Importantly, all the long-range interactions are supported by both 5′–3′ and 3′–5′ chimeras in our simplified SPLASH data (Supplementary Fig. 13), as well as in COMRADES data (Supplementary Fig. 14). The 3D structure

modeling of the FSE around region (12 K–15 K) also revealed the spatial proximity between arch 2 and Ziv's arch (Fig. 3d).

**Dynamics of RNA structure during viral life cycle of SARS-CoV-2.** Next, we analyzed interaction dynamics during phases of viral life cycle. A correlation analysis showed that samples from the same treatment group clustered together (Supplementary Fig. 15a). Principal Component Analysis (PCA) analysis on chimeric read counts indicates that virion RNA underwent major conformation alteration compared to RNAs in cell (C) and in

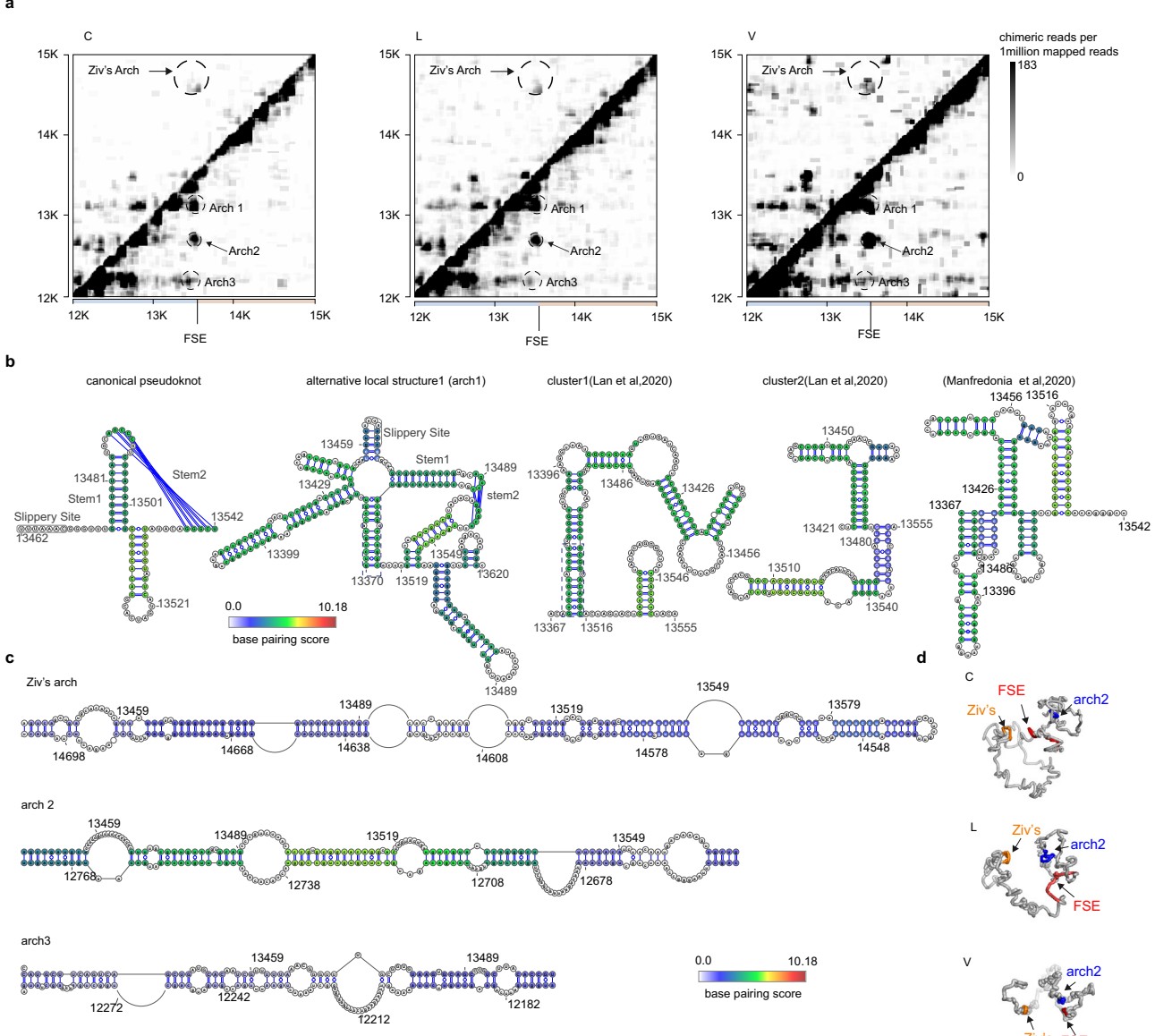

**Fig. 3 Alternative local and long-distance FSE structures. a** Heatmaps show chimeric reads spanning 12 K–15 K of SARS-CoV-2 genome in individual samples. Ziv's arch and alternative arches were plotted as indicated. **b** Base pairing of canonical pseudoknot and alternative structures as indicated, overlaid by base-pairing scores in C samples. **c** Base pairing of indicated arches. Colors represent the base-pairing score counts of non-redundant chimeric reads supporting each base pair. **d** 3D modeling of structure around FSE. FSE is in red, whereas nt14548–nt14708 (arch4 partner) is orange and nt12702–nt12802 is blue.

lysate (L), as shown along primary component 1 (Supplementary Fig. 15b).

Clustering of chimeras often break down with increasing sequencing depth, because separate clusters are merged together, and in extreme cases they can cover the entire transcript[21]. To overcome this issue, we decided to call enriched two-dimensional bin–bin interaction pairs by comparing ligated and corresponding non-ligated samples by DESeq2 as previously proposed[25].

In this case, we obtained 38,570 10 nt bin pairs enriched in ligated C samples, 65,405 pairs in L samples, and 114,677 pairs in V samples (logFoldChange > 1, false discovery rate (FDR) < 0.05). These interactions are provided as Supplementary Data 3 and are visualized as arc plot in Supplementary Fig. 16a. Most of the enriched interactions in C are also shared in the other two investigated stages, whereas in the meantime, there are many V sample-specific interactions (Supplementary Fig. 16b). At the

same time, we tested concordance between clusters from COMRADES data[16] and enriched interaction pairs from our simplified SPLASH data, and found gRNA clusters are also enriched in one group of our data set and, reciprocally, majority of the enriched interaction, especially in C group, overlap with gRNA interaction clusters from COMRADES data (Supplementary Fig. 16c).

Contrary to the relative short-range interactions of most enriched interaction pairs in C samples, more interactions in V samples are long range (Supplementary Fig. 16d). Most of the interactions are mediated by orf1ab and sgRNA regions, whereas <1% of enriched interactions are mediated by UTRs. Specifically, although V samples have much more enriched interactions, genome cyclization (interaction between 5′-UTR and 3′-UTR) and multiple sgRNA–sgRNA interactions are lost in V samples (Supplementary Fig. 16e).

To obtain better resolution of dynamics in RNA interactions in different stages. We then used DESeq2[25,39] to compare interaction strength in each $100\ nt \times 100\ nt$ window in the viral interaction map. After removal of the low-abundant pairs, pairwise comparisons between C, L, and V groups were made. Under a default cutoff ($log2FC > \pm 1$, $FDR < 0.05$), we found similar patterns of differential interactions in the comparisons of virions vs. cell (VvsC) and virion vs. lysate (VvsL) (Supplementary Fig. 17a), and fold changes of VvsC are correlated with both VvsL and LvsC (Supplementary Fig. 17b). This is in agreement with the closer relationship between C and L groups in PCA (Supplementary Fig. 15b), indicating the RNA conformation changes gradually from C to L and then to V.

A heatmap analysis of differential interactions (Fig. 4a) suggested a lower density of interactions in the 3′-third of SARS-CoV-2 genome in virions, compared to cells and lysates. Genome cyclization[25] was also reduced in virions, whereas proximal interactions and long-range interactions other than end-to-end cyclization were strengthened in virions (Fig. 4a). Differential interactions were classified into four categories: proximal (length ≤ 200 nt), proximal (length ≤ 200 nt), distal (5 kb < length ≤ 20 kb) and long range (20 kb < length ≤ 30 kb) according to length. Majority of differential interactions in the proximal and distal category are strengthened in virions (Fig. 4b).

An increase in proximal and distal interactions could also be observed in lysate and still visible when log2FC cutoff was elevated to 5 (Supplementary Fig. 17c). This is also concordant with more prominent distant interactions of virions (Fig. 1c and Supplementary Fig. 16a). These data might reflect compaction of genome during packaging into virions.

We then focused on the changes in interaction mediated by TRS-L region (first 100 nt). These interactions were weakened in virions, compared to cells or lysates, both for TRS-L-sgRNA long-range interaction and local folding at the 5′-end of ORF1ab. However, the TRS-L-S interaction was stronger in the lysate than in cells (Supplementary Fig. 17d), perhaps indicating the first steps of packaging of the viral genome.

Considering complicated sgRNAs derived from SARS-CoV-2, many of them are unannotated[3,40]. It is difficult to discriminate sgRNAs from fragmented gRNAs. Therefore, it is almost impossible to remove or separate all the sgRNAs from proximity data. Thus, the interactions we describe in this study can be derived from gRNAs or sgRNAs, especially for those overlapping to canonical sgRNAs. RNA-Seq coverage in the 3′-third of the genome, where canonical sgRNAs are located, is significantly higher than in the 5′ two-thirds. We observe that virions contain a higher coverage of interactions in the 5′ two-thirds of the genome, compared to cells and lysates (Fig. 4a). We suggest that this might be explained by virions being dominated by gRNA–gRNA interactions, whereas cells and lysates contain a higher fraction of sgRNA–sgRNA interactions (Fig. 4a and Supplementary Fig. 16e).

The RNA proximity ligation data heatmaps are often similar to mammalian genome Hi-C data[41,42] (Fig. 4c). This prompted us to check whether SARS-CoV-2 genome RNA are also compartmentalized into domains and whether global compaction of genome RNA results in impairment of domains. To this end, we applied an insulation score algorithm to call domain boundaries in SARS-CoV-2 genome[43]. In this way, SARS-CoV-2 genome was split into 76, 71, and 96 domains in C, L, and V samples, respectively. The average intra-domain contact matrices were shown as heatmap in Fig. 4d, indicating a reduction of inter-domain interactions. As expected, the insulation scores are significantly lower in boundaries (Fig. 4e). Remarkably, the insulation scores are highly correlated between groups of samples (Supplementary Fig. 18a) and domain boundaries are consistent in different

samples (Fig. 4f and Supplementary Fig. 18b). Concordantly, the domain length was comparable between samples (Supplementary Fig. 18c).

Furthermore, the boundary strength in V are significantly higher than in C and L samples (Fig. 4f), and ratio of intra-domain to inter-domain interactions was also higher in V samples (Supplementary Fig. 18d), indicating that during compaction and packaging of the genome, the domain structures were not only retained but even strengthened. Previous studies have revealed domains in Zika virus genome RNA[44] and compaction in virions[45]. Here we described that domains in SARS-CoV-2 are stably maintained during life cycle.

Finally, we calculated Shannon entropy values along the SARS-CoV-2 genome (Supplementary Fig. 19a). High entropy indicates flexible regions that may form multiple alternative base pairs[25]. As expected, the entropies are higher in cells than in virions, indicating that RNAs in cells adopt more flexible structures than in virions (Supplementary Fig. 19b). The entropies inversely correlate with insulation score (Supplementary Fig. 19c), indicating that domain boundaries are more flexible and might be more attractive sites for drug design. In addition, SARS-CoV-2 was proposed originating from recombination and the common evolutionary mechanism could lead to emerging human coronaviruses[46]. Seven putative breakpoints (at 99% confidence) were identified by comparing SARS-CoV-2 to five related coronavirus genomes (Supplementary Fig. 20). Strikingly, these putative breakpoints locate closely to domain boundaries. This shed light on a possibility that recombination of coronaviruses may be related to these domains, which would be worthy of investigating in the future when more evolutionary-related coronaviruses are found.

## Discussion

In this study, we developed a simplified SPLASH protocol based on proximity ligation, to capture RNA–RNA interactions in the SARS-CoV-2 virus. RNA proximity ligation has previously been used to address many questions in models ranging from viruses to animal tissues[18,21,33,34,36,42,44,47]. There are four major differences between simplified SPLASH protocol and traditional SPLASH as follows: (1) RNase III was used to fragment RNA as in PARIS[35] and COMRADES[16,25], this treatment makes all the fragmented RNA ends compatible for T4 RNA ligase, and this is supposed to increase efficiency of ligation; (2) T4 Polynucleotide Kinase treatment is omitted; (3) considering low input of RNA amount in virions, we decided to omit all the enrichment steps; and (4) the purified ligation products were directly subjected to a commercial pico-input strand-specific RNA-Seq library construction kit. The major purpose of these modifications is to reduce the complexity of the whole protocol to increase the RNA yield adequate for library construction, so that the protocol is more suitable for low amount of virion RNAs. As a result, we obtained 28.9% chimeric reads in virions and more than 10% chimeras in cells and lysates. The higher chimeric rates in virions might result from highly compact genome

Our results provide the first direct evidence that TRS-L regions form long-distance interactions with TRS-B regions. Although it was widely believed that long-range interactions are required for discontinuous transcription, direct experimental evidence for such interactions was lacking. By comparing TRS-L : TRS-B chimeras ligated in 3′–5′ and 5′–3′ orientations, we distinguished two classes of reads, which represented (1) RNA–RNA interactions and (2) sgRNAs.

We validated two of the putative sgRNAs by Sanger sequencing. Interestingly, although these novel sgRNAs do not have canonical cCS-B motif upstream of gene body, they both partially overlap with the canonical CS motif. This indicates that TRS-L

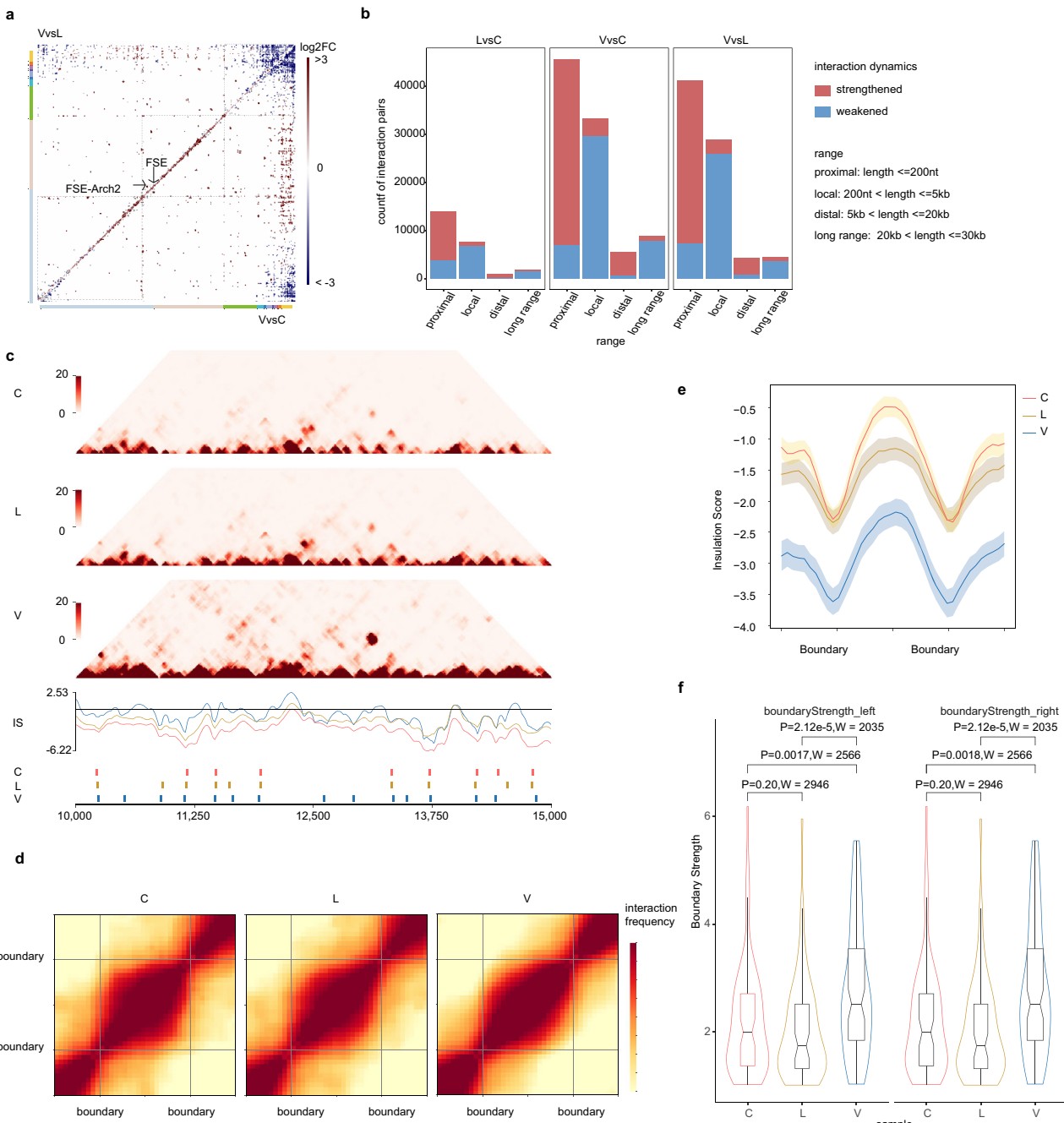

**Fig. 4 Dynamic structures in different phase of viral cycle. a** Heatmaps showing comparisons of RNA–RNA interactions in virions vs. cells (VvsC) and virions vs. lysates (VvsL). VvsL is in the upper quadrant and VvsC is in the lower quadrant. **b** Differential interactions were classified according to spans as indicated. Bar plots shows number of strengthened or weakened interactions in each category. **c** RNA interaction maps (Top) binned at 10 nt resolution show interactions 10–15 kb on SARS-CoV-2 genome in C, L, and V samples. Line plots (median) show insulation profiles. Short lines (bottom) reflect boundaries. **d** Maintenance of domains during SARS-CoV-2 virus life cycle. Heatmaps showing the normalized average interaction frequencies for all boundaries as well as their nearby regions (±0.5 domain length) in C, L, and V samples. The heatmaps were binned at 10 nt resolution. **e** The average normalized insulation scores were plotted around boundaries from 1/2 domain upstream to 1/2 domain downstream and divided into 40 bins. Shadowed area show SE bands at each bin. $n = 76, 71$, and 96 domains in C, L, and V samples. **f** Violin plot compare boundary strength among C, L, and V samples. Showing higher boundary strength in V samples. Boxplots show largest (upper whisker), smallest (lower whisker), 50% quantile (center), upper hinge (75% quantile), and lower hinge (25% quantile). Comparisons were made by two-sided two-sample Wilcoxon's rank-sum test, no multiple comparisons adjustment was made. P-value and Wilcoxon's rank-sum test statistics W were indicated. $n = 76, 71$, and 96 domains in C, L, and V samples.

and at least partial cCS-B base pairing at the negative strand are critical for template switching in discontinuous transcription of these sgRNAs. It would be interesting to further identify novel TRS-L-dependent sgRNAs and check if they play roles in SARS-CoV-2 biology.

The FSE structure has attracted attention, because it is of vital importance in translating nonstructural proteins in ORF1b and perturbing FSE has significance in modulating coronavirus[48]. A three-stemmed mRNA pseudoknot in the SARS coronavirus frameshift signal was proposed[10] to regulate this process and was

confirmed in SARS-CoV-2[11,38]. This structure is supported, as all the stems in the pseudoknot are covered by chimeras (Fig. 3b). The normalized base-pairing scores are higher in virions. In other cases, distinct structures other than the three-stem pseudoknot were reported[14,15]. In this study, we also proposed different structures around FSE, in which stems are also in the structures proposed in refs. [12,14]. At the same time, the alternative stems are also supported by chimeras; importantly, most stems in these structures are also supported by COMRADES data (Supplementary Fig. 11), suggesting FSE region is indeed highly dynamic and complicated as proposed previously. Interestingly, these stems also have more chimeras in virions (Supplementary Fig. 11), indicating that these structures are fine-tuned during its life cycle. Ziv et al.[16] found that the FSE of SARS-CoV-2 is embedded within a ~1.5 kb-long higher-order structure that bridges the 3′-end of ORF1a with the 5′-region of ORF1b, which was termed the FSE-arch. Here we speculate that Ziv's arch coexists with alternative structures, suggesting that regions around FSE are also dynamic, and that RNA conformation changes, presumably to fine tune frameshifting rates and stoichiometry of nonstructural proteins. These additional structures cooperate with Ziv's arch to embed the FSE in a larger "high-order pseudoknot." The large and small form of "pseudoknot" might provide a structural basis for ribosome stalling. The mechanisms balancing alternative structures, and transcription and translation of orf1a/b remain to be elucidated in the future. Interestingly, the alternative structures described above are also found in virions (Fig. 3a and Supplementary Fig. 11), suggesting that even packed into particles, the complicated conformations remained.

By comparing dynamics of SARS-CoV-2 during its life cycle, we found compaction of the SARS-CoV-2 genome in virions compared to cells (Fig. 4). We found more interactions, especially long-range ones, in V samples (Supplementary Fig. 16a, b, d). In addition, when we directly compare interaction strength of these bin pairs (Fig. 4), we found there are substantial long-range interactions, which are more prominent in V compared to C and L samples (Fig. 4a, b).

There are more long-range interactions in virions than in cells, possibly due to high compaction of genome in virions. Interestingly, not all the long-range interactions are strengthened in virions. We found that although genome is highly compacted, the head-to-tail genome cyclization were reduced in virions (Supplementary Figs. 6 and 17d). In terms of quantity, enriched interactions of 5′-UTR and 3′-UTR are also less in V samples (Supplementary Fig. 16e), which might facilitate the rapid initiation of virus life cycle after releasing into cells.

We found that the genome of SARS-CoV-2 is demarcated by domain boundaries, with intra-domain interactions stronger than inter-domain interactions. The uniform and regular domain folding is reminiscent of the nucleosomes-like, beads-on-string structure of eukaryotic DNA genome. Importantly, we also found that domains are stronger in virions and positions of domain boundaries remained consistent during life cycle, with a few domains merged in cells (Fig. 4f). The beads-on-string-like structure of SARS-CoV-2 ribonucleoprotein (RNP) in virion was also observed by cryo-electron microscopy technology[49]. Domains were also previously reported in the Zika virus[44]. We therefore speculate that domain organization is the rule rather than an exception of genome folding in single-strand RNA viruses. As boundaries remain stable in different phases of virus life cycle, we hypothesize that nucleocapsid (N) protein or other proteins involved in genome RNA packaging maintains its role as the regulator of the genome structure in infected cells, when the RNA is released. Interestingly, putative recombination breakpoints seem to locate closely to domain boundaries

(Supplementary Fig. 10). This make it reasonable to speculate that these domain boundaries are prone to recombination. Limited to lack of knowledge about the origin of this virus and the related viral genome, we are not able to completely describe landscape of reliable recombination breakpoints currently. In the future, it will be interesting to study the potential relationship between domain boundaries and recombination. To our knowledge, this is the first report on the dynamics and stability of domains in RNA virus genomes throughout its life cycle. It should be interesting to study whether other viruses have similar patterns in the genome structure during virus packaging and release. The delineation of structural elements underlying transcription regulation and other key viral replication processes might provide insights into designing better antiviral strategies.

## Methods

**Cell culture**. *Chlorocebus sabaeus* (Green monkey) VeroE6 (female, RRID:CV-CL_YQ49) were purchased from American Type Culture Collection (id: ATCC CRL-1586). VeroE6 cells were cultured in Dulbecco's modified Eagles medium supplemented with 10% fetal bovine serum at 37 °C in a humidified $CO_2$ incubator.

**Virus inoculation and crosslinking**. Infection experiments were performed under biosafety level 3 conditions. SARS-CoV-2 virus strain Wuhan-Hu-1 was kindly provided by (Wuhan Institute of Virology). Independent biological replicates were performed using 90–120 million cells each. VeroE6 cells were inoculated with SARS-CoV-2 strain Wuhan-Hu-1 at a multiplicity of infection = 0.01 pfu/cell for 24 h. Following inoculation, two flasks of cells were washed three times by phosphate-buffered saline (PBS) and then subjected to crosslinking. The remaining cells were cultured for another 48 h, when CPE was observed in about 70% cells, supernatant was collected and centrifuged at 4 °C 200 × g for 10 min to remove cell pellet. Then the clear supernatant was mixed with equal volume of saturated ammonium sulfate and incubate at 4 °C for 1 h. At the same time, the remaining unshed cells were washed three times by PBS and subjected to crosslinking.

For crosslinking, cells or virus pellet were incubated with 2 mM of EZ-Link Psoralen-PEG3- Biotin (ThermoFisher Scientific) at 37 °C for 10 min in PBS containing 0.01% digitonin. The cells were then spread onto a 10 cm plate and irradiated using 365 nm ultraviolet radiation for 20 min on ice. Cell and virion RNAs were extracted with RNeasy mini kit (Qiagen).

The virus inoculation and crosslinking were performed on three independent replicates.

**Simplified SPLASH assay**. Here, 500 ng of RNA was fragmented using RNase III (Ambion) in 20 μl mixture for 10 min at 37 °C and purified using 40 μl of MagicPure RNA Beads (TransGen). Each RNA sample was divided in two: one half was used for proximity ligation and then crosslink reversal (C, L, and V samples). Proximity ligation was done under the following conditions: 200 ng fragmented RNA, 1 unit/μl RNA ligase 1 (New England Biolabs), 1× RNA ligase buffer, 50 mM ATP, 1 unit/μl Superase-in (Invitrogen), final volume: 200 μl. Reactions were incubated for 16 h at 16 °C and were terminated by cleaning with miRNeasy kit (Qiagen). Crosslink reversal was done by irradiating the RNA on ice 254 nm UltraViolet C radiation for 5 min using a CL-1000 crosslinker (UVP). For the non-ligated controls, crosslink reversal was done directly after crosslinking, omitting proximity ligation (non-ligated C_N corresponding to C, L_N corresponding to L, and V_N corresponding to V).

**Sequencing library preparation**. Sequencing libraries were prepared with 50 ng input RNA material using SMARTer Stranded Total RNA-Seq Kit v2—Pico Input Mammalian (Takara Bio USA, Inc., USA), according to the manufacturer's instructions. The libraries were PCR amplified with 12 cycles using PCR primers in the kit and purified using 0.8× Ampure XP beads. The libraries were paired-end sequenced (PE150) using Illumina Nova seq platform.

**Data preprocessing**. Sequencing data were collected by Illumina CASAVA 1.8. We obtained an average of about 70 M raw paired-end reads from each replicates of samples. Data preprocessing was performed according to ref. [25]. In brief, raw paired-end reads were trimmed for adaptors and checked for quality using cutadapt (Martin, 2011)[50]. Chimeric reads were identified and annotated to the respective genome using hyb[51]. SARS-CoV-2 samples were processed using SARS-CoV-2 sequence (NC_045512.2 [https://www.ncbi.nlm.nih.gov/genome/?term=NC_045512.2]).

**Chimeras and interaction calling**. Chimeric reads were called and annotated with the hyb package (https://github.com/gkudla/hyb)[51], using the command:
default (stringent) parameters:
hyb analyse in=sample_R2.fq db=SARS-CoV-2_no_polyA Format=fastq align=bowtie2 eval=0.001

Relaxed pipeline:

hyb analyse in= sample_R2.fq db=SARS-CoV-2_no_polyA Format=fastq align=bowtie2 eval=0.001 gmax=20

The gmax setting represents the maximum gap/overlap between hybrid arms allowed when calling chimeras. Specifically, with the default setting (gmax = 4), hyb would only allow candidate chimeras where the two arms are either adjacent in the read (gap = 0), the mapped portions of the read are separated by a maximum of 4 nt (gap ≤ 4), or the mapped portions of the read overlap by a maximum of 4 nt (overlap ≤ 4). A stringent gmax setting (the default gmax = 4) reduces potential false-positive chimeras at the cost of increasing false-negative chimeras. In particular, we found that the default setting rejects chimeras that result from "spliced" TRS-L–TRS-B junctions, whereas the relaxed setting gmax = 20 allows the recovery of such chimeras.

PCR duplicates were collapsed as part of the preprocessing step of the hyb pipeline.

We also tested robustness of this pipeline by comparing read2 (sample_R2.fq) with pear merged PE reads:

pear -e -j 32 –f sample_R2.fastq.gz -r sample _R1.fastq.gz –o sample.PEAR

then detect chimeras using hyb:

hyb analyse in= sample.PEAR.assembled.fastq db=SARS-CoV-2_no_polyA Format=fastq align=bowtie2 eval=0.001

We found high similarity of contact matrices between the two pipelines (data are available upon request) and use R2 single reads (corresponding to RNA sense strand) hereafter, to make the analysis as simple as possible.

To evaluate the folding energy of chimeric reads, we used hybrid-min with default settings as in ref. [25]. We then randomly reassigned (shuffled) pairs of fragments found in chimeric reads and repeated the folding energy analysis. The folding energies of experimentally identified and shuffled chimeras were compared by Wilcoxon's test.

Virus interaction heatmaps were plotted using Java Treeview[52], as previously described[25], such that color intensity represents the coverage of chimeric reads at every pair of positions. The first read of each pair is plotted along the $X$ axis and the second read along the $Y$ axis. As a result, chimeras found in the 5′–3′ orientation are shown above the diagonal and chimeras in the 3′–5′ orientation are below the diagonal. Viewpoint histograms and arc plots were plotted with ggplot2 R package[53].

For TRS-L interaction peak calling, each chimeric read was split and mapped to two paired non-overlapping 10 nt bins; we first scored interactions by log2-transformed chimeric reads, then calculated $z$-scores for all the interaction pairs. The $z$-scores > 2.13 (which means log2 chimeras larger than average with 95% confidential) were considered as enriched interactions. Then, enriched TRS-L interaction were selected if either arm located in 1–100 nt.

The statistical significance of RNA–RNA interactions was calculated using DESeq2 as previously reported in ref. [25], by comparing counts of chimeric reads in 10 nt × 10 nt bin pairs from ligated and control data sets.

**RNA secondary structure folding and visualization.** For short-range interactions, we assembled non-zero chimeric groups into uninterrupted stem structures and then fold RNA using COMRADES (https://github.com/gkudla/comrades).

For long-range interactions, we first assembled uninterrupted stem structures as above for each arm and then fold RNA by hybrid-min in unfold-3.8[54].

Folded structures were visualized in VARNAv3-93. Paired bases are colored with normalized base-pairing scores as follows: scores were first calculated using COMRADES (https://github.com/gkudla/comrades) command as: comradesMakeConstraints -i Data.hyb -f genome.fasta -b 1 -e 29870. This score is defined as chimeric read count for each base pair. Then the scores were divided by total mapped read counts and log2 transformed to make scores comparable between samples. For a given folded structure, normalized base-pairing scores of each base pair in each sample were compared to non-ligated controls by paired Wilcoxon's rank-sum test. All the structures depicted in this study show significant ($P < 0.05$) enrichment of base pairing scores over non-ligated controls.

**ROC analysis.** The ROC approach was adopted to evaluate the performance of simplified SPLASH in detecting proximal RNA interactions. The consensus 5′-UTR structure was used as a golden standard. For each 5 nt bins, the true-positive data sets were defined by the consensus having base pairing between bases tested and the true-negative data sets were otherwise. To evaluate the simplified SPLASH data, we use base-pairing score to classify each base–base pair into a paired or unpaired group. The ROC curve was obtained by varying the threshold of the base-pairing score and counting the rates of true positives and false positives using colauc function in caTools R package[55].

**Calling of topological domain.** Domain boundaries were identified by insulation score[43] using the 10 nt resolution simplified SPLASH contact matrices data. Here we used 500 nt × 500 nt (50× resolution) square along each bin for calculating insulation score, A 50 nt (10× resolution) window was used for statistics of the delta vector and removed the weak boundaries, for which "boundary strength" < 1.

**Average insulation score of domain.** The average insulation scores were normalized by around all domain and their nearby regions (±0.5 domain length). The heatmaps were binned at 10 nt resolution and an 800 nt window. Average insulation scores were plotted around boundaries from 1/2 domain upstream to 1/2 domain downstream.

**Average interaction heatmap of domains.** The size of the domain was homogenized to 400 nt, and the upstream and downstream extended 1/2 domain calculated the interaction frequency by the averaging all domain. The resulted matrices were plotted as heatmap by log2 average signals.

**3D modeling of virus genome.** We used pastis-0.1.0[56] software to model RNA genome in three dimensions. The final results obtained by Multidimensional scaling (MDS) algorithm were used for 3D visualization. For the spatial location of particular gene loci, we used 1 point/20 balls to calculate the position of specific genes in the whole 3D simulation and then modify the pymol results by a python script.

**Recombination analyses.** Potential recombination events and the location of putative breakpoints in coronavirus genomes were detected using Simplot[57] and RDP[58]. Potential recombinant regions among analyzed sequences were identified by sliding a 200 nt window at a 10 nt step across the alignment using the Kimura two-parameter model. The positions of the analyzed sequence regions were based on those in the reference SARS-CoV-2 Wuhan-Hu-1 (MN908947). Regions between breakpoints were identified using a 99% confidence threshold.

**Differential interaction identifying.** Differential gene expression analysis was performed using DESeq2[39]. A 100 nt bin interaction that displayed more than ±1 log2FC (FDR < 0.01) between C or L and V samples were considered as significantly differential interactions. Then, log2FC heatmaps were plotted in using Java Treeview. The full script to identify differential interactions can be found at github (see below)

**RT-PCR of candidate novel sgRNAs.** RNA from infected non-crosslinked VERO cells (24 h, as described above) were extracted by miRNeasy Mini Kit (Qiagen). Then, 100 ng of RNA was subjected to retrotranscription using SuperScript VILO MasterMix (ThermoFisher) and cDNA was amplified with 2×Es Taq MasterMix (Cowin Biotech) by 0.4 μM of each primer: 3.9 K (5′-TGTTGTAACTTCTTCAA CACAAGC-3′) or 12.3 K (5′-TGTTCAAGGGAACACAACCATC-3′) and TRS-L (5′-CCCAGGTAACAAACCAACCAAC-3′).

**Statistics.** Statistical analyses for differential interaction was conducted with the R Bioconductor package DESeq2 using three independent replicates as described above.

Comparison of quantitative indicators, such as boundary strength, was performed with two-sided Wilcoxon's rank-sum test. For a given structure, to test whether the base pairing is enriched in ligated samples, or to compare base-pairing dynamics, we performed paired Wilcoxon's rank-sum test on normalized base-pairing scores.

To test whether differential interaction spannings follow the same continuous distribution, two-sided Kolmogorov–Smirnov test was performed.

Statistical significance of differences in odds ratios between two groups (Supplementary Fig. 6e, f) were calculated using a two-sided Fisher's exact test.

Correlation analysis of chimeric reads counts (Fig. 1b and Supplementary Fig. 1) and insulation scores (Supplementary Fig. 7a) between samples were performed by Pearson's product moment correlation coefficient (R or PCC).

All the statistics tests were performed with R package stats.

**Reporting summary.** Further information on research design is available in the Nature Research Reporting Summary linked to this article.

## Data availability
The data supporting the findings of this study are available from the corresponding authors upon reasonable request. The simplified SPLASH data generated in this study, along with Processed sequencing data sets analyzed in this study (hyb files), have been deposited in the Gene Expression Omnibus (GEO) database under accession code GSE164565. DESeq2 statistics for enrichment of interaction bin pairs in ligated samples are provided in Supplementary Data 3. The COMRADES data used in this study are available in the GEO database under accession GSE154662. The SARS-COV-2 reference genome can be found at https://www.ncbi.nlm.nih.gov/genome/?term=NC_045512.2.

## Code availability
Custom codes used for data analysis in this paper can be found at https://github.com/zany1983/simplifed_SPLASH (https://doi.org/10.5281/zenodo.5336972)[59].

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

## Acknowledgements
This study was supported by the National Science and Technology Major Projects (2018ZX10305410) to Z. Zhao, by the National Key Research and Development Program of China (2018YFA0900801) to Y.Z., and Wellcome grant 207507 to G.K.

## Author contributions
Z. Zhao conceived and supervised the project. G.K. supervised design and bioformatic analysis of the project. M.J. supervised virus infection and cell culture. Y.Z. performed simplified SPLASH experiments. K.H. and Z. Zou performed virus and cell culture, and crosslinking. D.X., J.Y.L., H.R., W.S., Y.M., and Y.W. performed bioinformatics analysis. D.W., S.S., and P.L. performed library construction and RT-PCR experiments. Y.Z., Z. Zhao, and G.K. wrote the manuscript. All authors critically revised the manuscript.

## Competing interests
The authors declare no competing interests.
