## [Peer Review File · Nature Communications]

In vivo structure and dynamics of SARS-CoV-2 RNA genomeREVIEWER COMMENTS

Reviewer #1 (Remarks to the Author):

The paper entitled "Insights into the secondary structural ensembles of the full SARS-CoV-2 RNA genome in infected cells" used a simplified SPLASH protocol to capture RNA-RNA interactions of SARS-CoV-2 virus, collected from 3 different samples (conditions/phases) in Vero cells. The authors identified interactions with alternative structures within the 5'-UTR, the 3'-UTR, the frameshifting element (FSE) pseudoknot, genomic cyclization, and the TRS-L and TRS-Bs. They further compared the interaction patterns for the three different samples and analyzed their dynamic changes. The study is timely and of high significance considering the COVID-19 pandemic. That being said, quite a few issues need to be addressed.

Major:

1. The biggest concern is about the technology and the data quality. This study is built on a new technology, a simplified version of SPLASH. Thus, more and careful quality controls are needed. Authors should provide details on the library construction, sequencing data, and intermediate results on the bioinformatics analysis. For example, how many PCR cycles are used in library construction? What about the PCR duplicates in the sequencing data and mapped reads (see point 6 below)? I noticed that the ratio of chimeric reads are very high in their sequencing results (lines 327-328), which should draw a lot of attention since, even with an enrichment step, normally PARIS and SPLASH have a very low ratio of chimeric reads. The authors should also compare the results of RNA interactions (intra- and inter-) with known ones (see the published PARIS and SPLASH papers). It should include not only examples of true positives, but also quantifications of the rates of false negatives and possible false positives. This reviewer also questions on Fig. 1b and Extended data Fig. 1. How the correlations are calculated? The mere correlation of number of chimeric reads is not enough. The agreement between different replicates should be quantified using the correlations between interaction matrices like in Fig. 1c.
2. Continue with the above point 1, more statistics and summaries of the results should be included to bring the readers more confidence in the technology and the data. How many interactions are finally obtained in total? How about their distributions? How many are in UTRs? What about the overlaps with Ziv et al. 2000? What about the agreements with other technologies, for example icSHAPE and other SHAPE-like probings (Sun et al. 2021, Huston et al. 2021) and computational prediction (Rangan et al. 2020)? Fig. 1c is very low resolution and not enough.
3. Fig 1c, the interactions within SARS-CoV-2 looks very evenly distributed without no much pattern, very different from previous studies of PARIS, SPLASH, and COMRADES on viruses. Why? Also, the virus in the virion (V) stage have more long interaction which is not consistent with the final conclusion that structure in virion are more local domains. Have the authors excluded or separated reads from subgenomic RNA? Most of the signals are in the subgenomic regions (also in Fig. 4).
4. The ligation and other related steps in the technology do not explain the big discrepancies between 3'-5' and 5'-3' chimeras. Why they are so different in Fig. 1d? Specially, why Ziv's Arch only has support of 5'-3' chimeric reads, Arch1 and 3 only have support of 3'-5' chimeric reads, but Arch2 has both?
5. Some disagreements between the description in the main text and associated figures raise big concerns. For example, around lines 132-133, it says "SL1~SL3 are stronger in cells than in virions", however it does not look true according to Extended Data Fig2a. Around lines 302-303: it says "As expected, the entropies are higher in virions than in cells, indicating that RNAs in cells adopt more flexible structures than in virions", which also seem problematic according to Extended Data Fig9b.
6. What is a "junction site" precisely defined? The gap within the two mapped fragments in a chimeric read, or collectively defined from a group of them? This reviewer is not clear on Fig. 2c. What a line represents? It lacks details on how a "junction site" is defined and obtained, and what it looks like.
7. The group of mapped reads have very sharp boundaries in Fig 2d. Are they simply PCR duplicates? And it is better to show the mapping of chimeric reads like Figure 2e for each of the stem in the structure models they present additional to the matrix plot, especially for the alternative models.

8. In Fig. 3. What exactly an "arch" means? The so-called arches should be labeled in Fig. 3b. Why Ziv's arch is an FSE-arch? It is not that close to FSE (Fig. 3d).
9. Fig. 4 is of very low resolution and thus this reviewer is not able to assess the results presented in almost the whole section of "Dynamics of RNA structure during viral life cycle of SARS-CoV-2".
10. The analysis of the domain demarcation looks interesting. But more details are needed. Do the domain boundaries change in C, L, and V samples? Are there any sequence patterns around the boundaries of the domains? If yes, do they conserve?
11. Some figures are not ordered the same way as they called in the main text.

Minor:

1. C_N/L_N/V_N has not been rigorously defined.
2. Lines 58~71, The sequence and structural elements are a lot, their relationships are complex, and the description is difficult to follow. It would be much better to have a schematic for this part.
3. is The Figure1b and Extended Data Fig1b are both showing the correlation for V group. Where is the one for C group?
4. Line 179, more details are need for "maximum gap/overlap setting". Here and below, parameters should be precisely defined and justified.
5. Line 237. In figure S6, the correlation of V1 and V2 is 1. Please check.
- 4 Line 326, "lib" should be "library".
4. Line 583. The author used hyb for data processing. Only R2 file of the paired files was used. Why not R1 or the merged file? An explanation should be provided.

Reviewer #2 (Remarks to the Author):

In this paper, Zhang and colleagues perform an analysis of the RNA structural dynamics of the SARS-CoV-2 virus genome, across different stages of the infection. Although several studies already performed a thorough analysis of SARS-CoV-2 RNA structures, no study so far has investigated how these structures dynamically change during the infection. This represents one of the key unanswered questions to date, and this article meets this need.

I have a few major questions/remarks for the authors (those concerning the FSE and 3'UTR pseudoknots are to me *crucial*):

- Page 5, line 124. The presence of chimaeras can also arise from template-switching activity of the reverse transcriptase. The authors should evaluate this possibility, as a possible source of artifacts, by checking whether the chimeras showing the strongest correlation are enriched for the GG dinucleotide (or GGG trinucleotide).

- Page 5, line 129. What about SL6, SL7 and the recently proposer SL8 (see Lan et al, 2020 and Manfredonia et al, 2020)? Are those supported as well?

- Page 5, line 140. "The pseudoknot in the 3' UTR had fewer chimeras". Nonetheless, from this sentence it seems like at least some chimeras exist that supported the presence of this pseudoknot. Is this correct? If this is the case, I would like the authors to discuss this in greater detail, as all the previous papers did not provide support for the existence of this pseudoknot (see Manfredonia et al, 2020; Ziv et al, 2020; Huston et al, 2020; etc.). This would be the first article to provide support for the existence of this pseudoknot, so the authors should discuss why this is the case. Also, does the abundance of the chimeras for the pseudoknot dynamically change across the analyzed infection

stages? This would provide experimental support to what has been hypothesized in a recent review (see doi: 10.1042/BST20200670).

- Page 5, line 141. The authors describe a potential alternative structure for the 3' UTR. In a recent report (see Morandi et al, 2021; doi: 10.1038/s41592-021-01075-w) it has been reported the existence of an alternative 3' UTR configuration, supported by both strong covariation and by COMRADES data (Ziv et al, 2020). Do the authors' data support this conformation as well?

- Page 6, line 156. Indeed, the presence of spliced viral RNAs can lead to artifacts in the analysis. This is why, Ziv and colleagues selectively pulled down the sgRNAs or the full length vRNA. Can the author use Ziv's data for the vRNA to provide orthogonal validation to the RNA-RNA interactions here reported?

- Page 6, lines 168-186. This part belongs more to the Online methods section, than to the results.

- Page 8, line 225. Once again, the authors are the first ones to provide support for the presence of the pseudoknotted structure at the level of the FSE. This should be thoroughly discussed. Again: does the abundance of the pseudoknot-supporting chimeras change across the infection time-points? Are the chimeras significantly enriched with respect to the non-ligated samples?

How do the authors explain this marked difference with other studies?

What about the alternative FSE structures reported by both Lan et al, 2020 and Morandi et al, 2021?

Are those supported?

Reviewer #1 (Remarks to the Author):

The paper entitled "Insights into the secondary structural ensembles of the full SARS-CoV-2 RNA genome in infected cells" used a simplified SPLASH protocol to capture RNA-RNA interactions of SARS-CoV-2 virus, collected from 3 different samples (conditions/phases) in Vero cells. The authors identified interactions with alternative structures within the 5'-UTR, the 3'-UTR, the frameshifting element (FSE) pseudoknot, genomic cyclization, and the TRS-L and TRS-Bs. They further compared the interaction patterns for the three different samples and analyzed their dynamic changes. The study is timely and of high significance considering the COVID-19 pandemic. That being said, quite a few issues need to be addressed.

Response:

We thank the reviewer for the summary of this work and kind words.

Major:

1. The biggest concern is about the technology and the data quality. This study is built on a new technology, a simplified version of SPLASH. Thus, more and careful quality controls are needed. Authors should provide details on the library construction, sequencing data, and intermediate results on the bioinformatics analysis. For example, how many PCR cycles are used in library construction? What about the PCR duplicates in the sequencing data and mapped reads (see point 6 below)? I noticed that the ratio of chimeric reads are very high in their sequencing results (lines 327-328), which should draw a lot of attention since, even with an enrichment step, normally PARIS and SPLASH have a very low ratio of chimeric reads. The authors should also compare the results of RNA interactions (intra- and inter-) with known ones (see the published PARIS and SPLASH papers). It should include not only examples of true positives, but also quantifications of the rates of false negatives and possible false positives. This reviewer also questions on Fig. 1b and Extended data Fig. 1. How the correlations are calculated? The mere correlation of number of chimeric reads is not enough. The agreement between different replicates should be quantified using the correlations between interaction matrices like in Fig. 1c.

1.1 The biggest concern is about the technology and the data quality. This study is built on a new technology, a simplified version of SPLASH. Thus, more and careful quality controls

are needed. Authors should provide details on the library construction, sequencing data, and intermediate results on the bioinformatics analysis. For example, how many PCR cycles are used in library construction?

Response:

Thank you for advices of quality controls, in the revised version, we have provided details on library construction (section methods, page 19-20), sequencing data (Supplementary Table1, methods, page 20), and intermediate results on the bioinformatics analysis (Supplementary data1 in page 5, Supplementary data2 in page 10). 12 cycles were used in library construction (section method, page 20)

The processed hyb files which are the most important intermediate files are submitted to GEO database (GSE164565) along with raw sequencing data. Then contact matrices and statistics for enrichment in ligated samples were submitted as Supplementary data1 and Supplementary data2, respectively.

1.2 What about the PCR duplicates in the sequencing data and mapped reads (see point 6 below)?

Response:

PCR duplicates (i.e. sequencing reads with identical sequence) were collapsed as part of the preprocessing step of the hyb pipeline. This is now clarified in the Methods section (page 21). In supplementary table 1, we could see PCR duplicates rates approximately 60%, which means 40% of mapped reads are non-redundant. This is comparable to COMRADES data (Supplementary Table 1).

1.3 I noticed that the ratio of chimeric reads are very high in their sequencing results (lines 327-328), which should draw a lot of attention since, even with an enrichment step, normally PARIS and SPLASH have a very low ratio of chimeric reads.

Response:

The chimeric rate varies among different methods. We noticed that chimeric rates in C and L samples are approximately 12%, which are in the range of popular proximity ligation assays (for example, MARIO has chimeric rates as high as 30% (Nguyen TC et al. 2016), and RIC-seq has about 15%(Cai Z et al., 2020). We noticed a higher chimeric rate in V samples, which might be results of high condensation of RNAs in virions. Even for the same method, chimeric rate varies depending on multiple factors including the efficiency of ligation,

sample types, chimera calling pipeline and sequencing read length (for example, in CLASH the chimeric rate ranged from less than 1% to more than 10% in reads of different lengths (Travis et al, 2014, Fig. 7). We speculate that the relatively high rate of chimeric reads detected in this study compared to the original SPLASH study (Aw et al., 2016) is due to a combination of mapping method (hyb vs BWA MEM in Aw et al.), filtering method (we retained short- and long-range interactions, whereas Aw et al. removed interactions spanning less than 50-nt), and our use of RNase III to fragmentize RNA, which produces RNA fragments compatible with T4 RNA ligase I, so that 100% of RNA fragment are suitable for ligating.

1.4 The authors should also compare the results of RNA interactions (intra- and inter-) with known ones (see the published PARIS and SPLASH papers). It should include not only examples of true positives, but also quantifications of the rates of false negatives and possible false positives. Response:

In this study, we focused on RNA structure of SARS-CoV-2 in vero cells and in virions. There is no golden standard of RNA structure in this circumstance yet. Instead, we have now compared the structures of 5'-UTR, 3'-UTR, and FSE in our data with the same structures published in previous reports (see new Supplementary Fig.4, Supplementary Fig.5 and Supplementary Fig.11, etc). The results show high consistency with existing reports, such as 5'-UTR, but also reveal new interactions, such as the interaction between the TRS-L and TRS-B sites. Given that 5'-UTR structure was reported to be similar in several recent publications, we used this structure as a gold standard to test sensitivity and specificity of our method to detect local structure. The ROC curve was obtained by varying the threshold of base pairing scores and counting the rates of true positives and false positives. The area under the curve is as high as 0.97 for our simplified SPLASH data, indicating high specificity and sensitivity of our methods to detect RNA structure. We have included this analysis in the results section (page 5, and Supplementary Fig4e and 4f).

1.5 This reviewer also questions on Fig. 1b and Extended data Fig. 1. How the correlations are calculated? The mere correlation of number of chimeric reads is not enough. The agreement between different replicates should be quantified using the correlations between interaction matrices like in Fig. 1c.

Response:

Yes, this is precisely how these correlations were calculated. Specifically, correlations are calculated based on chimeric read count of pairs of 10nt x 10nt bins between two samples. Points shown in Fig.1 b and Supplementary Fig.2 shows pairwise correlation between replicates, and Supplementary Fig.15 shows Pearson's correlation coefficients between any two samples. Additionally, we added contact matrices from replicates of samples, to show the raw contact frequencies in Supplementary Fig.1.

2. Continue with the above point 1, more statistics and summaries of the results should be included to bring the readers more confidence in the technology and the data. How many interactions are finally obtained in total? How about their distributions? How many are in UTRs? What about the overlaps with Ziv et al. 2000? What about the agreements with other technologies, for example icSHAPE and other SHAPE-like probings (Sun et al. 2021, Huston et al. 2021) and computational prediction (Rangan et al. 2020)? Fig. 1c is very low resolution and not enough.

2.1 Continue with the above point 1, more statistics and summaries of the results should be included to bring the readers more confidence in the technology and the data. How many interactions are finally obtained in total?

Response:

To quantify interactions, we have now used DESeq2 to call interactions enriched in simplified SPLASH data, relative to non-crosslinked data. This resembles the approach used in the original COMRADES paper (Ziv O, et al, Nat Methods. 2018). After multiple testing correction, this identified 38570, 65405 and 114677 ($\log_2(\text{FoldChange}) > 1$, $\text{FDR} < 0.05$) 10nt x 10nt bins pairs with significantly enriched chimeras in C, L, and V samples respectively. We have included this analysis in page 10.

2.2 How about their distributions? How many are in UTRs?

Response:

We found majority of interactions are involved in ORF regions, with less than 1% of these enriched interactions are in UTRs. This is summarized in results section (pages 11). The length and coordination distribution of these interactions are summarized in Supplementary Fig.16d and 16e. We described these results in page 11.

2.3 What about the overlaps with Ziv et al. 2000?

Response:

We have extensively compared the interactions described by Ziv et al., 2020. These results are summarized in the text (page 11) and Supplementary Fig.16c.

2.4 What about the agreements with other technologies, for example icSHAPE and other SHAPE-like probings (Sun et al. 2021, Huston et al. 2021) and computational prediction (Rangan et al. 2020)?

Response:

When we study specific structures, for examples, 5'-UTR, FSE and 3'-UTR, we compared our results with other technologies. Generally, these structures are comprehensively consistent with other technologies, for examples, we identified all the well characterized stem-loops in 5'-UTR. Importantly, although computer prediction suggested a small stem-loop after SL4 in 5'-UTR, our data didn't support this stem-loop, consistent to most icSHAPE and related methods (Sun et al, 2021, Huston, et al, 2021, Manfredonia, et al, 2020, Manfredonia, et al, 2021, Lan, et al. 2020). We also compared multiple alternative structures in FSE and 3'-UTRs, and found cross-validation of some unique structures which are only partially consistent with some studies. We have extensively compared and discussed in results sections. (For examples, page5, page 6 and page 9).

2.5 Fig. 1c is very low resolution and not enough.

Response:

We have improved the resolution of Figure 1C and now includes zoomed-in insets to illustrate the resolution of the underlying data in selected regions.

3. Fig 1c, the interactions within SARS-CoV-2 looks very evenly distributed without no much pattern, very different from previous studies of PARIS, SPLASH, and COMRADES on viruses. Why? Also, the virus in the virion (V) stage have more long interaction which is not consistent with the final conclusion that structure in virion are more local domains. Have the authors excluded or separated reads from subgenomic RNA? Most of the signals are in the subgenomic regions (also in Fig. 4).

3.1 Fig 1c, the interactions within SARS-CoV-2 looks very evenly distributed without no much pattern, very different from previous studies of PARIS, SPLASH, and COMRADES on viruses. Why?

Response:

This pattern is mostly caused by the large size of the SARS-CoV-2 genome and small size of the Figure, which caused local interactions to appear as single pixels in the previous Figure 1C. To prove that the interactions are not in fact evenly distributed, we provide zoomed-in insets in Figure 1C and new Supplementary Figures 1-2. We also provide additional heatmaps below, to show contact matrix in the first 5Kb region.

Additionally, we could observe in Fig.4c that the genome is organized in domains. This pattern is similar to previous reports (such as Fig.2 in Cai Z, et al. Nature. 2020 and Fig.3 in Li P, et al. Cell Host Microbe. 2018.).

3.2 Also, the virus in the virion (V) stage have more long interaction which is not consistent with the final conclusion that structure in virion are more local domains.

Response:

According to reviewer's suggestion, we quantified enriched interactions in each stage, and found more enriched 10nt×10nt bins pairs of interaction in V samples. And especially more

long range interactions in V samples (Supplementary Fig.16b and 16d). In addition, when we directly compare interaction strength of these bin pairs (Figure 4a and 4b), we found there are substantial long range interactions which are more prominent in V compared to C and L samples (Figure 4a and 4b).

There are more long range interactions in virions than in cells, possibly due to high compaction of genome in virions. At the same time, when we look into RNAs in virions, we could also see more chimeric reads intra domain than inter-domain (Supplementary Fig.18c). These results are not contradictory, but rather two sides of the same coin. The RNA is highly condensed and demarcated by domains in virions, then the beaded like domains are packed in the lumen, rendering more long range interactions than in cells. We proposed a model of RNA conformation in cells and in virions below.

Interestingly, not all the long-range interactions are strengthened in virions. We found that although genome is highly compacted, the head-to-tail genome cyclization were reduced in virions(Supplementary Fig6, Supplementary Fig.17d). Quantatively, enriched interaction of 5'-UTR and 3'-UTR are also less in V samples (Supplementary Fig.16e). These data suggest that genome cyclization might be involved in virus packaging or replication. We have added this discussion in discussion section (page 15-16).

3.3 Have the authors excluded or separated reads from subgenomic RNA? Most of the signals are in the subgenomic regions (also in Fig. 4).

Response:

We identified three characteristics that differentiated sgRNA chimeras from bona-fide RNA-RNA interaction chimeras: (1) sgRNAs were ligated almost exclusively in the 5'-3' orientation, whereas RNA-RNA interaction chimeras can be ligated in both 5'-3' and 3'-5' orientation (reviewed in ref21); (2) the junctions between arms of chimeras were precisely localised in sgRNAs, whereas ligation sites in RNA-RNA interaction chimeras were variable, due to the random nuclease digestion step used in proximity ligation 44; (3) sgRNA chimeras typically included regions of homology between TRS-L and TRS-B sides of the

chimera 46, whereas RNA-RNA interaction chimeras typically include no such regions. Adjustment of the maximum gap/overlap setting in our analysis pipeline, *hyb*, allows detection ($g_{max}=20$, "relaxed pipeline", allowing a maximum of 20nt gap or overlap between chimera fragments) or removal ($g_{max}=4$, "stringent pipeline", see methods section for details) of most sgRNA chimeras, while RNA-RNA interaction chimeras are detected with both settings. We therefore used the stringent pipeline to analyse chimeras. We describe this important point in the results section (pages 7-8).

On the other hand, it is difficult to discriminate RNA-RNA interactions in subgenomic RNAs from fragmented gRNAs, especially considering complicated subgenomic RNAs derived from SARS-CoV-2, many of them are unannotated. Therefore, it was impossible to remove or separate all the subgenomic RNA at least in our technology. We now clarify in the results section (page 12) that the interactions we describe can be derived from gRNAs or subgenomic RNAs.

4. The ligation and other related steps in the technology do not explain the big discrepancies between 3'-5' and 5'-3' chimeras. Why they are so different in Fig. 1d? Specially, why Ziv's Arch only has support of 5'-3' chimeric reads, Arch1 and 3 only have support of 3'-5' chimeric reads, but Arch2 has both?

4.1 The ligation and other related steps in the technology do not explain the big discrepancies between 3'-5' and 5'-3' chimeras. Why they are so different in Fig. 1d?

Response:

In our opinion, the theoretic ratio of read counts of 3'-5' and 5'-3' chimeras is not 1:1.

As shown above, RNase III produce RNA fragments with partially based pairs, as illustrated in the figure above, both of stem-loops and stems could be proximity ligated. The stem-loop form circles, which could only be detected as 3'-5' chimeras (because sequenced RNA in the other orientation is exactly same to original RNA, and could not be identified as chimera). While stems result in both 3'-5' and 5'-3' chimeras, however short-range 5'-3' chimeras are more difficult to detect because *hyb*, as well as other chimera mapping tools,

identify short distance 5'-3' chimeras as internal deletions. So, there are more 3'-5' chimeras than 5'-3' chimeras in a typical proximity ligation data. This is especially true when interactions are local. For example, we classify all the both simplified SPLASH data in this study and COMRADES data (from O Ziv et.al, 2020) chimeras by the distances between two arms, and found in short distance chimeras (distance < 200nt), 3'-5' chimeras are dominant, while in long distance chimeras (distance >200nt), 3'-5' and 5'-3' chimeras are comparable. We have added this analysis in page 5.

4.2 Specially, why Ziv's Arch only has support of 5' -3' chimeric reads, Arch1 and 3 only have support of 3' -5' chimeric reads, but Arch2 has both?

Response:

All of these arches have both 5'-3' and 3'-5' chimeric reads (Supplementary Fig.13). But the Reviewer is right that arches have biased orientation of chimeric reads. It's difficult to explain why one type of chimeric reads are more prominent (refer review in Kudla G, et.al. Annu Rev Genomics Hum Genet. 2020). We speculate that this might be partially explained by relative low coverage of chimeric reads in Ziv's Arch and Arches 1 and 3.

5. Some disagreements between the description in the main text and associated figures raise big concerns. For example, around lines 132-133, it says "SL1~SL3 are stronger in cells than in virions", however it does not look true according to Extended Data Fig2a. Around lines 302-303: it says "As expected, the entropies are higher in virions than in cells, indicating that RNAs in cells adopt more flexible structures than in virions", which also seem problematic according to Extended Data Fig9b.

Response:

We are terribly sorry for writing errors. SL1~SL3 are weaker in cells than in virions. And the entropies are higher in cells than in virion. we have corrected the mistakes in the revised manuscript. We clarified these in page 6 and page 13 respectively.

6. What is a "junction site" precisely defined? The gap within the two mapped fragments in a chimeric read, or collectively defined from a group of them? This reviewer is not clear on Fig. 2c. What a line represents? It lacks details on how a "junction site" is defined and obtained, and what it looks like.

Response:

Junction sites are defined as ligation points of the two fragments. So, either of end of segment1 (e1) or start site of segment2 (s2) is the so-called junction site. We made an illustration of junction sites in Supplementary Fig.7c. These junction sites are in single nucleotide resolution. To analyze distribution of junction sites localised within first 100nt, for each nucleotide in this range, we count chimeras that either junction site is on this nucleotide. We now clarify this in Supplementary Fig.7c. So the lines in Fig. 2c and Supplementary Fig. 7d shows count of chimeras, whose junction sites are in every particular sites around TRS-L region (first 100nt). The plots show the distribution junction sites from stringent pipeline (Fig.2c) are not specific to nt75, which is the breakpoint of TRS-L dependent transcripts, while junction sites of relax chimeras, which are majorly sgRNAs are highly specific to nt75 (Supplementary Fig. 7d). So chimeras in stringent/default pipeline should indicate information of long range interaction of TRSL-TRSBs, rather than simply discontinuous transcripts.

7. The group of mapped reads have very sharp boundaries in Fig 2d. Are they simply PCR duplicates? And it is better to show the mapping of chimeric reads like Figure 2e for each of the stem in the structure models they present additional to the matrix plot, especially for the alternative models.

Response:

All the PCR duplicates are collapsed in the hyb pipeline, therefore, the sharp boundaries in Fig.2d are not due to PCR duplicates. As a matter of fact, the matrix was binned with window size of 10nt, so all the chimeras with end points that fall into the same 10nt-bins looked identical. More detailed chimeric information is in Fig.2e, from which we can see all the chimera arms are not the same, although they have the hotspots.

In the revised manuscript, we included details of chimeric reads in the Supplementary Fig. 8, Supplementary Fig.13 and Supplementary Fig.14 to show mapping details of chimeric reads supporting long range interactions.

8. In Fig. 3. What exactly an "arch" means? The so-called arches should be labeled in Fig. 3b. Why Ziv's arch is an FSE-arch? It is not that close to FSE (Fig. 3d).

Response:

Omer Ziv et al found FSE of SARS-CoV-2 is embedded within a much larger high order structure that bridges the 3' end of ORF1a with the region of ORF1b, which they termed the FSE- arch (Ziv, O., et al, 2020). In this study, we found apart from the FSE- arch, there are additional long range interactions, so we also term these arches after Omer Ziv et al. The arch1 is labeled in the alternative structure of FSE in Fig.3. Thank you for your reminding.

9. Fig. 4 is of very low resolution and thus this reviewer is not able to assess the results presented in almost the whole section of "Dynamics of RNA structure during viral life cycle of SARS-CoV-2".

Response:

We are terribly sorry for the low-resolution figure, we have changed this to a better resolution figure.

10. The analysis of the domain demarcation looks interesting. But more details are needed. Do the domain boundaries change in C, L, and V samples? Are there any sequence patterns around the boundaries of the domains? If yes, do they conserve?

Response:

Yes, domain boundaries are not exactly the same in C, L, and V samples, but the insulation score is correlated between these samples. We added overlapping analyzes on boundaries and found more than a half of boundaries are remained during the stages analyzed (Supplementary Fig. 18b).

We also performed motif analysis on sequence around boundaries, and found no motif is enriched, probably the domain and boundaries are not formed because of specific sequences.

11. Some figures are not ordered the same way as they called in the main text.

Response :

Sorry. we have carefully revised the manuscript, and they are in the same order now.

Minor:

1. C_N/L_N/V_N has not been rigorously defined.

Response:

Sorry, C_N/L_N/V_N are non-ligated controls for C, L and V samples, we have amended manuscript (page 20) accordingly.

2. Lines 58~71, The sequence and structural elements are a lot, their relationships are complex, and the description is difficult to follow. It would be much better to have a schematic for this part.

Response:

These elements are important for forming high-order structures promoting discontinuous RNA synthesis during N sgRNA transcription in the TGEV coronavirus. However, these elements are not conserved in beta-coronaviruses, so it's not possible to label in this manuscript, but we refer to a review describing the model in page 3.

3. is The Figure1b and Extended Data Fig1b are both showing the correlation for V group. Where is the one for C group?

Response:

We are sorry for the mistake, and we have change the C group in the Supplementary Fig.2.

4. Line 179, more details are need for "maximum gap/overlap setting". Here and below, parameters should be precisely defined and justified.

Response :

This refers to the gmax setting in the hyb pipeline. The gmax setting represents the maximum gap/overlap between hybrid arms allowed when calling chimeras. Specifically, with the default setting (gmax=4), hyb would only allow candidate chimeras where the two arms are either adjacent in the read (gap=0), the mapped portions of the read are separated by a maximum of 4 nt (gap<=4), or the mapped portions of the read overlap by a maximum of 4 nt (overlap <=4), this default setting was determined to be optimal for detecting interactions(Travis AJ, et al., 2014). A stringent gmax setting (such as the default gmax=4) reduces potential false positive chimeras at the cost of potentially increasing false negative chimeras. In particular, we found that the default setting rejects chimeras that result from "spliced" TRS-L - TRS-B junctions, whereas the relaxed setting gmax=20 allows the recovery of such chimeras. We now explain this in the Methods section (Page 22).

5. Line 237. In figure S6, the correlation of V1 and V2 is 1. Please check.

Response:

Thank you for your reminding. The correlation of V1 and V2 are calculated as 0.9968, as we used default decimal digits=2, so it was approximated to 1, We have fixed this issue and changed this figure accordingly.

4 Line 326, "lib" should be "library".

Response:

Thank you for correcting. We have corrected this mistake.

4. Line 583. The author used hyb for data processing. Only R2 file of the paired files was used. Why not R1 or the merged file? An explanation should be provided.

Response:

In fact, we also used pear software to merge paired end reads, and then performed downstream analysis. And found they are almost the same as to contact matrix and correlation of samples. Blow bar plot shows correlation between the two pipelines in each sample, and heatmaps show contact frequencies from the pear merged reads in each replicated. For simplicity, we used only R2 (RNA sense strand) reads for further analysis.

Reviewer #2 (Remarks to the Author):

In this paper, Zhang and colleagues perform an analysis of the RNA structural dynamics of the SARS-CoV-2 virus genome, across different stages of the infection. Although several studies already performed a thorough analysis of SARS-CoV-2 RNA structures, no study so

far has investigated how these structures dynamically change during the infection. This represents one of the key unanswered questions to date, and this article meets this need.

Response:

We would like to thank the reviewer for the points.

I have a few major questions/remarks for the authors (those concerning the FSE and 3'UTR pseudoknots are to me *crucial*):

- Page 5, line 124. The presence of chimaeras can also arise from template-switching activity of the reverse transcriptase. The authors should evaluate this possibility, as a possible source of artifacts, by checking whether the chimeras showing the strongest correlation are enriched for the GG dinucleotide (or GGG trinucleotide).

Response:

Thank you for your reminding, we also agree template switching might also contribute to chimeric reads. Accordingly, we have performed motif analysis on ends of top 5% chimeras, and found no motifs were enriched.

Then we counted top 20 5-nucleotides that are at the ends of chimeras, and plotted the chimeras counts (see the above bar plots), we didn't find enrichment of GG or GGG in these chimeras. We now state this in the manuscript (page 5).

- Page 5, line 129. What about SL6, SL7 and the recently proposed SL8 (see Lan et al, 2020 and Manfredonia et al, 2020)? Are those supported as well?

Response:

Yes, they are supported. We extended 5'-UTR region to 1-500nt, and found additional stem-loops, which are supportive to SL6, SL7, and pSL8 (Supplementary Fig.4). We added this information in page 5.

- Page 5, line 140. "The pseudoknot in the 3' UTR had fewer chimeras". Nonetheless, from this sentence it seems like at least some chimeras exist that supported the presence of this pseudoknot. Is this correct? If this is the case, I would like the authors to discuss this in greater detail, as all the previous papers did not provide support for the existence of this pseudoknot (see Manfredonia et al, 2020; Ziv et al, 2020; Huston et al, 2020; etc.). This would be the first article to provide support for the existence of this pseudoknot, so the authors should discuss why this is the case. Also, does the abundance of the chimeras for the pseudoknot dynamically change across the analyzed infection stages? This would provide experimental support to what has been hypothesized in a recent review (see doi: 10.1042/BST20200670).

Response:

Thank you for this comment. We now provide an analysis of simplified SPLASH support for the pseudoknot structure (see Supplementary Fig 5). In our opinion, the pseudoknot is also supported by COMRADES data (blue lines in Fig5A in O Ziv et al, 2020 and Supplementary Fig.5e), but with less chimeras, probably because of relative low coverage. We didn't find significant changes of base pairing score between cells and virion. Of note, the pseudoknot is also supported by NMR and 3D models recently (Wacker A, et al, Nucleic Acids Res, 2020. Rangan R, et al. Nucleic Acids Res. 2021). However, due to the relative low coverage, we speculate that this pseudoknot is not predominant. We made revisions in page 6.

- Page 5, line 141. The authors describe a potential alternative structure for the 3' UTR. In a recent report (see Morandi et al, 2021; doi: 10.1038/s41592-021-01075-w) it has been reported the existence of an alternative 3' UTR configuration, supported by both strong covariation and by COMRADES data (Ziv et al, 2020). Do the authors' data support this conformation as well?

Response:

Thank you for this reminding. We have evaluated this conformation by counting chimeras supporting base pairings (base pairing scores, method section in page 22). As results, all of the stems in conformation A, which is almost canonical structure, are supported. While for conformation B, four stems (including BSL and P2) are supported, however two short stems have no chimeras. This result might support that conformation A is a more dominant conformation. We think that this result is in agree with de convolution results in the recent paper (Morandi et al, 2021; doi: 10.1038/s41592-021-01075-w). We have included this analysis in page 6.

- Page 6, line 156. Indeed, the presence of spliced viral RNAs can lead to artifacts in the analysis. This is why, Ziv and colleagues selectively pulled down the sgRNAs or the full length vRNA. Can the author use Ziv's data for the vRNA to provide orthogonal validation to the RNA-RNA interactions here reported?

Response :

Thank you for this comment. To validate the interactions we report, we calculated a ROC curve which shows sensitivity and specificity of detecting local structure (page 5, Supplementary Fig.4e, 4f). In addition, we performed extensive comparison of our data with Ziv et al data (shown in the Supplementary Fig.8, 11, 14 and etc., which overall shows good agreement between the datasets). At the same time, we agree that it is difficult to distinguish gRNA from sgRNA interactions in our data, and therefore we now edited the relevant paragraph (page 12) to avoid over interpreting differences between gRNA-mediated and sgRNA-mediated interactions.

- Page 6, lines 168-186. This part belongs more to the Online methods section, than to the results.

Response :

Thank you for this comment, but we respectfully disagree with this opinion. This is a novel result that describes, for the first time, the differences between chimeras found in proximity ligation and RNA-Seq experiments and we believe that it is important enough to highlight this in the results section. In addition, this result is essential for justifying the type of filtering we use in subsequent stages of the analysis.

We also added more detailed information of the parameter in stringent and relax pipeline in the methods discussion (page 21).

- Page 8, line 225. Once again, the authors are the first ones to provide support for the presence of the pseudoknotted structure at the level of the FSE. This should be thoroughly discussed. Again: does the abundance of the pseudoknot-supporting chimeras change across the infection time-points? Are the chimeras significantly enriched with respect to the non-ligated samples?

How do the authors explain this marked difference with other studies?

Response:

Yes, we found chimeras in our data supporting pseudoknotted structure at the level of the FSE. We have rechecked the data, and found chimeras supporting base pairing of the pseudoknot both in cell and in virions, with higher base pairing scores in virions (Supplementary Fig.11a). More importantly, we use the same pipeline to analysis Ziv's COMRADES data, and found most stems in the canonical pseudoknot or alternative structures are also supported by COMRADES data. Some stems such as stem1 in the canonical pseudoknot are not covered by chimeras, might due to relative low coverage in COMRADES data. More importantly, the FSE pseudoknot was also captured by cryo-EM and functional analysis (Bhatt PR, et al. Science. 2021). Therefore, we think it's of high confidence that this pseudoknot exists. We have included this in pages 9-10 and Supplementary Fig.11 in the revised manuscript.

For all the base pairings structures, we have compared base pairing scores of each structure between ligated and non-ligated samples by wilcox test, all the structures depicted in this paper show significant higher base pairing scores in ligated samples. We described this in the methods section in page 22.

What about the alternative FSE structures reported by both Lan et al, 2020 and Morandi et al, 2021? Are those supported?

Response :

Thanks for reminding, we have checked these structures, and find stems are all supported by chimeras (page 15, and Supplementary Fig.11). Interestingly, we found most of these stems are also supported by Ziv's COMRADES data, suggesting that the alternative structures proposed around FSE exist, and FSE structures are indeed dynamic and complicated.

REVIEWERS' COMMENTS

Reviewer #1 (Remarks to the Author):

I am very happy to see the overall much improved work, with now adequate details on the methodology and comparisons to other-related studies. All my questions and concerns have been addressed.

Reviewer #2 (Remarks to the Author):

The authors have addressed all my points. I believe that this paper provides a unifying view on the complexity of the structure of the SARS-CoV-2 genome, bridging the gap between the findings of different groups, and as such, I strongly recommend it for publication in Nat. Comm.